# Six-month post-intensive care outcomes during high and low bed occupancy due to the COVID-19 pandemic: A multicenter prospective cohort study

Ana Castro-Avila[1,2]*, Catalina Merino-Osorio[1], Felipe González-Seguel[1,3]*, Agustín Camus-Molina[1,3], Felipe Muñoz-Muñoz[4], Jaime Leppe[1], on behalf of the IMPACCT COVID-19 study group[¶]

1 Carrera de Kinesiología, Facultad de Medicina, Clínica Alemana Universidad del Desarrollo, Santiago, Chile, 2 Department of Health Sciences, University of York, Heslington, United Kingdom, 3 Servicio de Medicina Física y Rehabilitación, Departamento de Medicina Interna, Facultad de Medicina, Clínica Alemana Universidad del Desarrollo, Santiago, Chile, 4 Centro de Paciente Crítico Adulto, Clínica INDISA, Santiago, Chile

¶ Members of the IMPACCT COVID-19 study group are provided in the Acknowledgments.
* feligonzalezs@udd.cl (FGS); anacastro@udd.cl (ACA)

**Data Availability Statement:** Email for data access: ceccasudd@udd.cl.

## Abstract

### Introduction

The COVID-19 pandemic can be seen as a natural experiment to test how bed occupancy affects post-intensive care unit (ICU) patient's functional outcomes. To compare by bed occupancy the frequency of mental, physical, and cognitive impairments in patients admitted to ICU during the COVID-19 pandemic.

### Methods

Prospective cohort of adults mechanically ventilated >48 hours in 19 ICUs from seven Chilean public and private hospitals. Ninety percent of nationwide beds occupied was the cut-off for low versus high bed occupancy. At ICU discharge, 3- and 6-month follow-up, we assessed disability using the World Health Organization Disability Assessment Schedule 2.0. Quality of life, mental, physical, and cognitive outcomes were also evaluated following the core outcome set for acute respiratory failure.

### Results

We enrolled 252 participants, 103 (41%) during low and 149 (59%) during high bed occupancy. Patients treated during high occupancy were younger ($P_{50}$ [$P_{25}$-$P_{75}$]: 55 [44–63] vs 61 [51–71]; p<0.001), more likely to be admitted due to COVID-19 (126 [85%] vs 65 [63%]; p<0.001), and have higher education qualification (94 [63%] vs 48 [47%]; p = 0.03). No differences were found in the frequency of at least one mental, physical or cognitive impairment by bed occupancy at ICU discharge (low vs high: 93% vs 91%; p = 0.6), 3-month (74% vs 63%; p = 0.2) and 6-month (57% vs 57%; p = 0.9) follow-up.

**Funding:** IMPACCT COVID-19 study was funded by Universidad del Desarrollo (Grant number 2020-78) and sponsored by the Chilean National Agency for Research and Development (ANID-0772). The funders had no role in the design, collection, analysis of the study, and writing of this manuscript. Received by CM-O.

**Competing interests:** The authors have declared that no competing interests exist.

## Conclusions

There were no differences in post-ICU outcomes between high and low bed occupancy. Most patients (>90%) had at least one mental, physical or cognitive impairment at ICU discharge, which remained high at 6-month follow-up (57%).

## Clinical trial registration

NCT04979897 (clinicaltrials.gov).

## Introduction

The COVID-19 pandemic challenged health systems globally. One challenge was the rapid increase in demand for critical care, which produced a large cohort of intensive care unit (ICU) survivors. Patients discharged from ICUs can present with various short- and long-term impairments collectively known as Post-Intensive Care Syndrome (PICS) [1]. Before the pandemic, 56% of ICU survivors had at least one mental, physical or cognitive impairment one year after discharge [2, 3]. For ICU patients admitted with COVID-19, rates of new disabilities, cognitive impairment, and symptoms of anxiety/depression seem to be similar to patients admitted due to other diagnoses [4, 5].

The rapid nature of the increase in demand for ICU beds during the pandemic might have led to delays in admissions [6] or shortcomings in the infrastructure available for delivering care. Australia, Ireland, and the United Kingdom recommend that acute care hospitals operate at an average of 85% bed occupancy [7–10] and not exceed 90% to deliver safe care [7]. The rationale for this recommendation is twofold: 1) higher bed occupancy leads to a greater workload for healthcare professionals, which is associated with increased mortality [8], and 2) reduced spare bed capacity can lead to delays in admission, which is associated with increased mortality [9–11]. Lower quality of care [12] and increased risk of readmission [13] have also been associated with higher bed occupancy.

To date, hospital bed occupancy above 85–90% has been associated with waiting times in Emergency departments [14], readmissions [13], hospital-acquired infections, and mortality [15]; however, its relationship with patients' post-intensive care functional outcomes remains unclear. The COVID-19 pandemic created conditions where ICUs were forced to operate above the recommended 90% capacity, providing a natural experiment for exploring the relationship between occupancy and outcomes. The primary aim of this study was to compare the frequency of mental, physical, and cognitive impairments at ICU discharge, 3 and 6 months of patients discharged from ICU who were admitted during high and low bed occupancy periods during the COVID-19 pandemic. The secondary aims were to compare mental, physical, and cognitive impairments of patients admitted to ICU due to COVID-19 versus other diagnoses at the same time points and to determine the survival 22 months after ICU discharge by bed-occupancy level and COVID-19 infection status.

## Material and methods

'Prospective, multicenter, cohort study recruiting patients between October 12[th], 2020, and April 10[th], 2021, at 19 ICUs in seven Chilean public and private hospitals. The study protocol was published [16] and registered on ClinicalTrials.gov (NCT04979897). This study was performed in line with the principles of the Declaration of Helsinki. Ethical approval was granted by the Clínica Alemana Research and Clinical Trials Unit, Faculty of Medicine, Clínica

Alemana Universidad del Desarrollo Ethics Committee (registration number 2020–78) and the Servicio de Salud Metropolitano Oriente Ethics Committee (registration number 152–0029).

## Patients

Adult patients (≥18 years old) on mechanical ventilation (MV)>48 hours in ICU due to any diagnoses. Exclusion criteria were patients who were uncooperative [17] or with delirium [18]; admitted due to severe burns, trauma or neurological disorder; were unable to walk independently two weeks prior to admission, understand or speak Spanish, or communicate verbally (details in the study protocol [16]). All included patients signed a written informed consent after receiving verbal information about the study.

## Patient subgroups

A threshold of 90% of national staffed ICU beds occupied (S1 Fig) was used to define low (i.e., October 12th, 2020–January 6th, 2021) and high bed occupancy (i.e., January 7th, 2021–April 10th, 2021). Additionally, we defined two groups based on the COVID-19 infection status (i.e., positive or negative laboratory polymerase chain reaction test).

## Data collection

The site coordinator screened patients daily to identify those within 72 hours of being discharged from ICU, inviting those eligible to participate. Once the patient consented, we collected from their clinical records: age, gender, body mass index, admission diagnosis, Charlson Comorbidity Index score, duration of MV, length of hospital and ICU stay, number of intubations, and the maximum level of organ system support received [19].

## ICU discharge and post-intensive care outcomes

The core outcome set for acute respiratory failure [20] was used to assess post-ICU outcomes (S1 Table). Experienced researchers (CM-O, AC-M, and FG-S) delivered a 3-hour training session to physiotherapists from the participating sites to standardize the assessment. At ICU discharge, evaluators assessed frailty [21], peripheral muscle strength [22], mobility [23], and cognitive function [24]. Immediately after, patients reported their disability [25], symptoms of depression, anxiety [26] and post-traumatic stress [27], their educational level, and employment status [28]. At the 3- and 6-month follow-up, three interviewers contacted patients via email or telephone to evaluate their cognitive function, disability, symptoms of depression, anxiety and post-traumatic stress, health-related quality of life, and employment status [20].

We retrieved death certificates from the National Civil Registry before the 3- and 6-month follow-up, and on August 15th, 2022, using the patient's national identification number, completing 22 months of follow-up for the first enrolled patient.

## Statistical analysis

For each subgroup and time point, categorical variables are presented as absolute and relative frequencies and differences tested for significance using chi-square or Fisher's exact. All continuous variables and the questionnaire scores did not follow a normal distribution; therefore, we used median ($P_{25}$- $P_{75}$) and Kruskal-Wallis test for significance.

The scores for each outcome were modelled using longitudinal linear multilevel regression with robust standard errors to account for clustering (i.e., seven sites), including an interaction term between time and occupancy level. Survival was analyzed using Cox regression models. All models were adjusted for age, sex, educational level, and COVID-19 infection. We

compared the sociodemographic and clinical characteristics of loss-to-follow-up patients and those assessed to test for the missing-at-random assumption. The Bonferroni correction for multiple testing was used to adjust p-values. All analyses were performed in Stata 16.0 SE.

## Results

Between October 12th, 2020, and April 10th, 2021, we screened 1,317 patients across 19 ICUs from seven participating sites. Of the 404 patients who met inclusion criteria, 252 were enrolled and evaluated at ICU discharge (Fig 1): 103 (41%) during low bed occupancy and 149 (59%) during high bed occupancy.

The sociodemographic and clinical features of the cohort are presented in Table 1, while functional outcomes are in S2 Table. Patients assessed at all stages were not significantly different from those lost to follow-up regarding the main outcomes on discharge (S3 and S4 Tables).

### High vs low bed occupancy

Patients admitted in the high occupancy period were younger (median[p25-p75]: 55[44–63] vs 61[51–71]; p-value<0.001), more likely to be employed full-time (n[%]: 107[82%] vs 40[39%]; p-value<0.001), more likely to be admitted due to COVID-19 infection (126[85%] vs 65[63%]; p-value<0.001), less likely to be classified as frail (2[1%] vs 11[11%]; p-value<0.001), more likely to have a higher education qualification (94[63%] vs 48[47%]; p-value = 0.03), spent fewer days on MV (8[6–13] vs 11[6–22] days; p-value = 0.009), and in the ICU (13[9–19] vs 18 [12–34] days; p-value<0.001).

### COVID-19 vs non-COVID-19 admissions

Patients admitted with COVID-19 infection were more likely to be male (137[71%] vs 26 [43%]; p-value<0.001), less likely to be classified as frail (7[3.6%] vs 6 [10%]; p-value = 0.05), have a lower Charlson Comorbidity Score (0[0–1] vs 1[0–3]; p-value<0.001); have a higher education qualification (120[63%] vs 22[37%]; p-value<0.001), and spent longer on MV (9[6–16.5] vs 8[4–13]; p-value = 0.01), and in the ICU (15[11–27] vs 13[8–20.5]; p-value = 0.036).

### Post-ICU patient's functional outcomes

Overall, at ICU discharge, 92% (95% CI: 88 to 95) of patients had at least one mental, physical, or cognitive impairment, which decreased to 68% (58 to 77) at the 3-month and to 57% (44 to 69) at the 6-month follow-up. Six months after ICU discharge, 39% (27 to 51) patients had mental impairments, 12% (5.3 to 22) had mental and cognitive impairments, and cognitive impairments or severe disability and mental impairments were each present in 3% (0.4 to 10).

Clinical frailty was present in 5% (2.8 to 8.7) of patients, while intensive care unit-acquired weakness (ICU-AW) was found in 34% (28 to 40) at ICU discharge and was similar across groups (Table 1). Cognitive impairments had a steady decrease over the follow-up period (72% [66 to 77] vs 36% [27 to 46] vs 16% [8.5 to 27]).

At ICU discharge, anxiety and post-traumatic stress symptoms were at 58% (51 to 64) and 65% (59 to 71), respectively. The frequency of both decreased by the 3-month follow-up; however, these remained at around 36% for the 3- and 6-month follow-ups. In the case of depression symptoms, 29% (24 to 35) had them at ICU discharge, which increased to 42% (33 to 52) at the 3-month follow-up and remaining at 39% (27 to 52) at the six months (Table 2).

According to the WHODAS 2.0 standardized disability level, severe or moderate disability was present in 54% (48 to 61) of the sample at ICU discharge, and this steadily decreased over time (30% [22 to 40] at three and 19% [11 to 31] at six months). The utility score in the EQ-

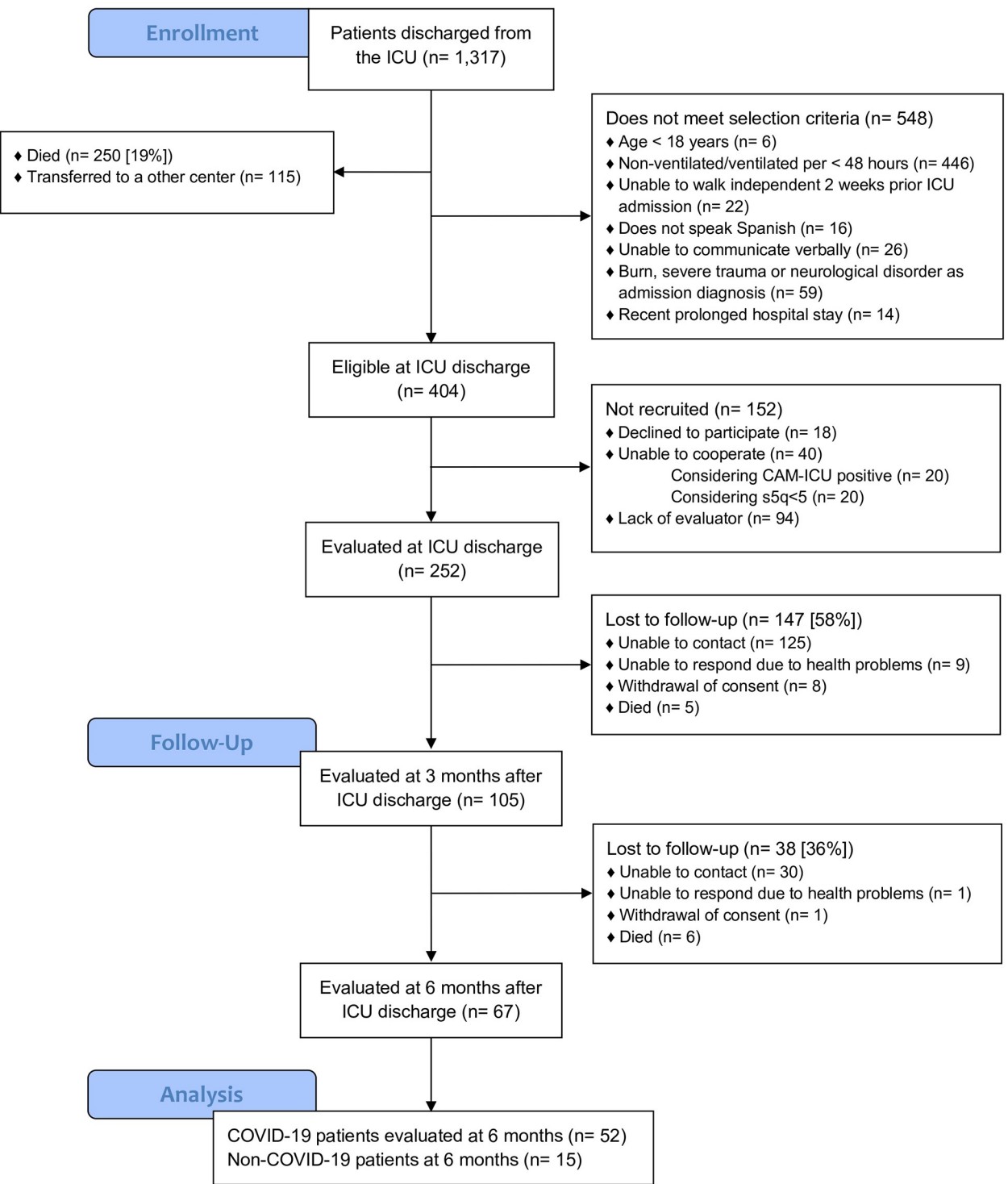

**Fig 1. STROBE flow chart of eligible patients.**

**Table 1. Baseline and clinical characteristics of the included patients.**

| | | Non–COVID-19 | | | COVID-19 | | |
| --- | --- | --- | --- | --- | --- | --- | --- |
| | Overall (*n* = 252) | Low bed occupancy (*n* = 37) | High bed occupancy (*n* = 23) | *P-*value | Low bed occupancy (*n* = 66) | High bed occupancy (*n* = 126) | *P*-value |
| Age, years | 57 (46.5–67) | 58 (44–69) | 56 (40–64) | 0.62 | 62 (55–71) | 55 (44–62) | <0.001 |
| Female sex | 89 (35.3%) | 20 (54.1%) | 14 (60.9%) | 0.6 | 21 (31.8%) | 34 (27.0%) | 0.48 |
| Body mass index, kg/m2 [a] | 29 (26–33.2) | 27.7 (25–34.8) | 29.5 (25–42.3) | 0.44 | 30 (27–32) | 29 (26.9–33.3) | 0.88 |
| Educational level | | | | 0.65 | | | 0.25 |
| <9 years | 42 (16.7%) | 14 (37.8%) | 8 (34.8%) | | 8 (12.1%) | 12 (9.5%) | |
| 9 to 12 years | 68 (27.0%) | 11 (29.7%) | 5 (21.7%) | | 22 (33.3%) | 30 (23.8%) | |
| >12 years | 142 (56.3%) | 12 (32.4%) | 10 (43.5%) | | 36 (54.5%) | 84 (66.7%) | |
| Baseline employment status | | | | 0.63 | | | <0.001 |
| Employed–Full Time | 147 (58.3%) | 11 (29.7%) | 9 (39.1%) | | 29 (43.9%) | 98 (77.8%) | |
| Employed–Part Time | 30 (11.9%) | 6 (16.2%) | 4 (17.4%) | | 10 (15.2%) | 10 (7.9%) | |
| Unemployed | 37 (14.7%) | 10 (27.0%) | 3 (13.0%) | | 16 (24.2%) | 8 (6.3%) | |
| Retired | 38 (15.1%) | 10 (27.0%) | 7 (30.4%) | | 11 (16.7%) | 10 (7.9%) | |
| Clinical Frailty Scale | 3 (2–3) | 3 (2–3) | 3 (2–4) | 0.64 | 3 (2–3) | 2 (2–3) | 0.006 |
| Very fit | 36 (14.3%) | 5 (13.5%) | 2 (8.7%) | 0.65 | 8 (12.1%) | 21 (16.7%) | 0.017 |
| Well | 84 (33.3%) | 9 (24.3%) | 5 (21.7%) | | 18 (27.3%) | 52 (41.3%) | |
| Managing well | 97 (38.5%) | 15 (40.5%) | 10 (43.5%) | | 27 (40.9%) | 45 (35.7%) | |
| Vulnerable | 22 (8.7%) | 3 (8.1%) | 5 (21.7%) | | 7 (10.6%) | 7 (5.6%) | |
| Mildly frail | 5 (2.0%) | 2 (5.4%) | 0 (0.0%) | | 2 (3.0%) | 1 (0.8%) | |
| Moderately frail | 7 (2.8%) | 2 (5.4%) | 1 (4.3%) | | 4 (6.1%) | 0 (0.0%) | |
| Severely frail | 1 (0.4%) | 1 (2.7%) | 0 (0.0%) | | 0 (0.0%) | 0 (0.0%) | |
| Charlson Comorbidity Index | 0 (0–1) | 2 (0–3) | 1 (0–3) | 0.17 | 0 (0–1) | 0 (0–1) | 0.69 |
| Admission diagnosis | | | | 0.38 | | | 0.17 |
| Non–COVID-19 pneumonia or ARDS | 16 (6.3%) | 8 (21.6%) | 8 (34.8%) | | 0 (0.0%) | 0 (0.0%) | |
| COVID-19 pneumonia or ARDS | 191 (75.8%) | 0 (0.0%) | 0 (0.0%) | | 65 (98.5%) | 126 (100.0%) | |
| Abdominal surgery | 10 (4.0%) | 7 (18.9%) | 3 (13.0%) | | 0 (0.0%) | 0 (0.0%) | |
| Heart failure | 9 (3.6%) | 6 (16.2%) | 3 (13.0%) | | 0 (0.0%) | 0 (0.0%) | |
| Septic shock | 14 (5.6%) | 6 (16.2%) | 7 (30.4%) | | 1 (1.5%) | 0 (0.0%) | |
| Drug intoxication/ suicide attempt | 3 (1.2%) | 3 (8.1%) | 0 (0.0%) | | 0 (0.0%) | 0 (0.0%) | |
| Cardiac arrest | 3 (1.2%) | 3 (8.1%) | 0 (0.0%) | | 0 (0.0%) | 0 (0.0%) | |
| Other | 6 (2.4%) | 4 (10.8%) | 2 (8.7%) | | 0 (0.0%) | 0 (0.0%) | |
| Organ System Supported during ICU stay | | | | | | | |
| Advanced Respiratory Support | 252 (100.0%) | 37 (100.0%) | 23 (100.0%) | | 66 (100.0%) | 126 (100.0%) | |
| Basic Cardiovascular Support | 158 (62.7%) | 19 (51.4%) | 16 (69.6%) | 0.16 | 39 (59.1%) | 84 (66.7%) | 0.3 |
| Advanced Cardiovascular Support | 84 (33.3%) | 18 (48.6%) | 7 (30.4%) | 0.16 | 20 (30.3%) | 39 (31.0%) | 0.93 |
| Renal Support | 16 (6.3%) | 8 (21.6%) | 2 (8.7%) | 0.19 | 4 (6.1%) | 2 (1.6%) | 0.09 |
| Liver Support | 1 (0.4%) | 0 (0.0%) | 0 (0.0%) | | 0 (0.0%) | 1 (0.8%) | 0.47 |
| Reintubated during ICU stay | 32 (12.7%) | 8 (21.6%) | 0 (0.0%) | 0.02 | 10 (15.2%) | 14 (11.1%) | 0.49 |
| Duration of ventilation, days | 9 (6–15.5) | 8 (4–13) | 7 (4–13) | 0.76 | 13 (8–34) | 8 (6–13) | <0.001 |
| ICU length of stay, days | 15 (10–25) | 15 (9–20) | 10 (8–21) | 0.2 | 21.5 (13–42) | 13 (10–19) | <0.001 |
| MRC-SS at ICU discharge | 52 (45–57) | 52 (44–57) | 50 (43–56) | 0.37 | 48 (44–55) | 52.5 (46–58) | 0.01 |
| ICU-AW | | | | 0.27 | | | 0.025 |

*(Continued)*

**Table 1.** (Continued)

| | Overall (*n* = 252) | Non–COVID-19 | | | COVID-19 | | |
| | | Low bed occupancy (*n* = 37) | High bed occupancy (*n* = 23) | *P-value* | Low bed occupancy (*n* = 66) | High bed occupancy (*n* = 126) | *P-value* |
|---|---|---|---|---|---|---|---|
| Without ICU-AW (MRC-SS ≥48) | 167 (66.3%) | 25 (67.6%) | 15 (65.2%) | | 37 (56.1%) | 90 (71.4%) | |
| Significant ICU-AW (MRC-SS 36–47) | 71 (28.2%) | 12 (32.4%) | 6 (26.1%) | | 21 (31.8%) | 32 (25.4%) | |
| Severe ICU-AW (MRC-SS <36) | 14 (5.6%) | 0 (0.0%) | 2 (8.7%) | | 8 (12.1%) | 4 (3.2%) | |
| FSS-ICU at ICU discharge | 26 (20–32) | 27 (19–31) | 25 (15–29) | 0.48 | 26.5 (19–32) | 26.5 (21–32) | 0.45 |
| Inability to walk [b] | 78 (31%) | 13 (35.1%) | 7 (30.4%) | 0.78 | 23 (34.8%) | 35 (28%) | 0.41 |

Definition of abbreviations: COVID-19 = coronavirus disease; ICU = intensive care unit; ARDS = acute respiratory distress syndrome; CFS = Clinical Frailty Scale; MRC-SS = Medical Research Council Sum Score; ICU-AW = Intensive Care Unit Acquired Weakness; FSS-ICU = Functional Status Score for the Intensive Care Unit.

Data are median (quartile 1–quartile 3) or n (%). Percentages may not total 100 because of overlaying or rounding.

[a] Denotes different sample size for non–COVID-19 low bed occupancy (n = 19), COVID-19 low bed occupancy (n = 41), non–COVID-19 high bed occupancy (n = 14), and COVID-19 high bed occupancy (n = 88).

[b] Defined as FSS-ICU walking item <2 points.

5D-3L was 0.70 (0.56–0.8) at three months and 0.78 (0.56–1) at six months. The proportion of patients reporting problems in the five domains of the questionnaire remained similar at the 3- and 6-month follow-up.

At three months post ICU discharge, 68% (58 to 76) of patients had changed their occupation or working arrangements. While 38% (27 to 50) had stopped working or studying, 49% (37 to 61) had reduced weekly working hours (Table 2). At six months, 61% (49 to 73) had changed their occupation or working arrangements.

## Post-ICU outcomes by bed occupancy and COVID-19 infection

Despite the clinical and demographic differences between patients admitted in the high vs low bed occupancy period, we did not find differences in the frequency of at least one mental, physical or cognitive impairment at ICU discharge (low vs high: 93% [95% CI: 86 to 97] vs 91% [95% CI: 86 to 95]), 3-month (74% [95% CI: 60 to 85] vs 63% [95% CI: 49 to 76]) and 6-month follow-up (57% [95% CI: 39 to 73] vs 57% [95% CI: 37 to 74]). The same applies to patients admitted due to COVID-19 vs other non-COVID-19 diagnoses at ICU discharge (COVID-19 vs non-COVID-19: 91% [95% CI: 86 to 94] vs 97% [95% CI: 88 to 99]), 3-month (70% [95% CI: 58 to 79] vs 64% [95% CI: 41 to 83]) and 6-month follow-up (58% [95% CI: 43 to 71] vs 53% [95% CI: 27 to 79]) (S5–S8 Tables). Our longitudinal adjusted models confirmed this finding (Fig 2), except for cognitive function at the 3-month follow-up, where patients admitted during the high-occupancy period have higher scores in the MoCA-Blind, but they are as likely to have cognitive impairment.

Between October 10[th], 2020 and August 15[th], 2022, 16 patients died after ICU discharge, which equates to a mortality rate (95% confidence interval) of 6.3% (3.7 to 10) or 4.2 (2.6 to 6.9) deaths per person-year. In a model adjusted by age, sex and educational level, survival was similar by bed occupancy, but patients infected with COVID-19 had better survival compared to patients admitted due to other diagnoses (Hazard Ratio [95% confidence interval]: 0.09 [0.03 to 0.27]; p-value<0.0001) (Fig 3).

**Table 2. Functional outcomes at intensive care unit discharge, 3- and 6-month follow-up according to bed occupancy.**

| | ICU discharge | | | 3-month follow-up | | | 6-month follow-up | | |
|---|---|---|---|---|---|---|---|---|---|
| | Low bed occupancy (*n* = 103) | High bed occupancy (*n* = 149) | *p-value* | Low bed occupancy (*n* = 50) | High bed occupancy (*n* = 55) | *p-value* | Low bed occupancy (*n* = 37) | High bed occupancy (*n* = 30) | *p-value* |
| WHODAS–Standardized disability level, % | 30 (11–56) | 27 (10–43) | 0.10 | 12 (5–26) | 10 (3.6–29) | 0.87 | 13.5 (2–23) | 5 (2–18) | 0.26 |
| WHODAS–Total score | 76 (48–109) | 70 (50–99) | 0.24 | 40 (31–53) | 44 (28–55) | 0.73 | 38 (28–49) | 35 (31–44) | 0.53 |
| Understanding & communicating | 25 (4–46) | 17 (4–37.5) | 0.28 | 8 (0–20.8) | 8 (4–37.5) | 0.17 | 12.5 (0–29) | 10 (0–29) | 0.96 |
| Mobility | 30 (5–80) | 25 (0–50) | 0.07 | 10 (0–40) | 15 (0–40) | 0.90 | 5 (0–30) | 2.5 (0–25) | 0.55 |
| Self-Care | 12.5 (0–56) | 6.3 (0–44) | 0.40 | 0 (0–12.5) | 0 (0–12.5) | 0.77 | 0 (0–6) | 0 (0–0) | 0.03 |
| Getting along with people | 20 (0–40) | 10 (0–30) | 0.05 | 0 (0–15) | 5 (0–10) | 0.62 | 0 (0–10) | 0 (0–10) | 0.72 |
| Life Activities: household | 19 (0–75) | 12.5 (0–56) | 0.44 | 6.3 (0–25) | 6.3 (0–25) | 0.83 | 12.5 (0–31) | 0 (0–19) | 0.04 |
| Life Activities: work or school | 12.5 (0–75) | 12.5 (0–63) | 0.58 | 12.5 (0–25) | 25 (0–50) | 0.21 | 25 (0–31) | 0 (0–0) | 0.01 |
| Participation in society | 44 (22–69) | 34 (16–56) | 0.07 | 16 (16–41) | 25 (16–38) | 0.36 | 25 (6–38) | 6 (0–18) | 0.04 |
| WHODAS–Level of disability | | | 0.27 | | | 0.96 | | | 0.79 |
| No disability (<5%) | 14 (13.6%) | 23 (15.4%) | | 16 (32%) | 17 (31%) | | 13 (35%) | 14 (47%) | |
| Mild disability (5–24%) | 29 (28.2%) | 49 (32.9%) | | 20 (40%) | 20 (36%) | | 16 (43%) | 11 (37%) | |
| Moderate disability (25–49%) | 28 (27.2%) | 47 (31.5%) | | 12 (24%) | 15 (27%) | | 7 (19%) | 4 (13%) | |
| Severe disability (50–95%) | 32 (31.1%) | 30 (20.1%) | | 2 (4%) | 3 (5%) | | 1 (3%) | 1 (3%) | |
| MoCA–Blind | 16 (11–18) | 16 (12–18) | 0.29 | 18 (15–20) | 20 (17–21) | 0.03 | 20 (18–22) | 21 (19–22) | 0.11 |
| Cognitive impairment (<18) | 75 (72.8%) | 106 (71.1%) | 0.77 | 22 (44%) | 16 (29%) | 0.11 | 6 (16%) | 5 (17%) | 0.96 |
| HADS–depression score | 5 (2–9) | 5 (2–8) | 0.28 | 7 (5–10) | 6 (5–10) | 0.35 | 7 (5–9) | 7 (5–8) | 0.48 |
| Normal (0–7) | 69 (67.0%) | 109 (73.2%) | 0.37 | 27 (54%) | 33 (61%) | 0.37 | 20 (54%) | 21 (70%) | 0.41 |
| Borderline abnormal (8–10) | 15 (14.6%) | 22 (14.8%) | | 14 (28%) | 9 (17%) | | 10 (27%) | 5 (17%) | |
| Abnormal (>11) | 19 (18.4%) | 18 (12.1%) | | 9 (18%) | 12 (22%) | | 7 (19%) | 4 (13%) | |
| HADS–anxiety score | 9 (5–12) | 8 (5–12) | 0.98 | 6 (4–9) | 6 (5–10) | 0.63 | 7 (3–10) | 6 (3–9) | 0.62 |
| Normal (0–7) | 42 (40.8%) | 65 (43.6%) | 0.63 | 31 (62%) | 34 (63%) | 0.98 | 24 (65%) | 21 (70%) | 0.76 |
| Borderline abnormal (8–10) | 23 (22.3%) | 26 (17.4%) | | 9 (18%) | 10 (19%) | | 6 (16%) | 3 (10%) | |
| Abnormal (>11) | 38 (36.9%) | 58 (38.9%) | | 10 (20%) | 10 (19%) | | 7 (19%) | 6 (20%) | |
| IES-R | 45 (27–57) | 42 (25–55) | 0.35 | 20 (7–39) | 21 (10–41) | 0.47 | 22 (7–44) | 13.5 (5–38) | 0.55 |
| Normal (0–23) | 20 (19.4%) | 32 (21.5%) | 0.65 | 27 (55%) | 29 (53%) | 0.98 | 20 (54%) | 17 (57%) | 0.63 |
| Some PTSD symptoms (24–32) | 14 (13.6%) | 22 (14.8%) | | 5 (10%) | 7 (13%) | | 2 (5%) | 4 (13%) | |
| Likely diagnosis of PTSD (33–36) | 6 (5.8%) | 14 (9.4%) | | 4 (8%) | 5 (9%) | | 3 (8%) | 2 (7%) | |
| PTSD (>36) | 63 (61.2%) | 81 (54.4%) | | 13 (27%) | 14 (25%) | | 12 (32%) | 7 (23%) | |

Definition of abbreviations: WHODAS = WHO Disability Assessment Schedule; MoCA-blind = Montreal Cognitive Assessment-blind; HADS = Hospital Anxiety and Depression Scale; IES-R = Impact of Event Scale-Revised; PTSD = Post-Traumatic Stress Disorder.

Data are median (quartile 1–quartile 3) or n (%). Percentages may not total 100 because of rounding.

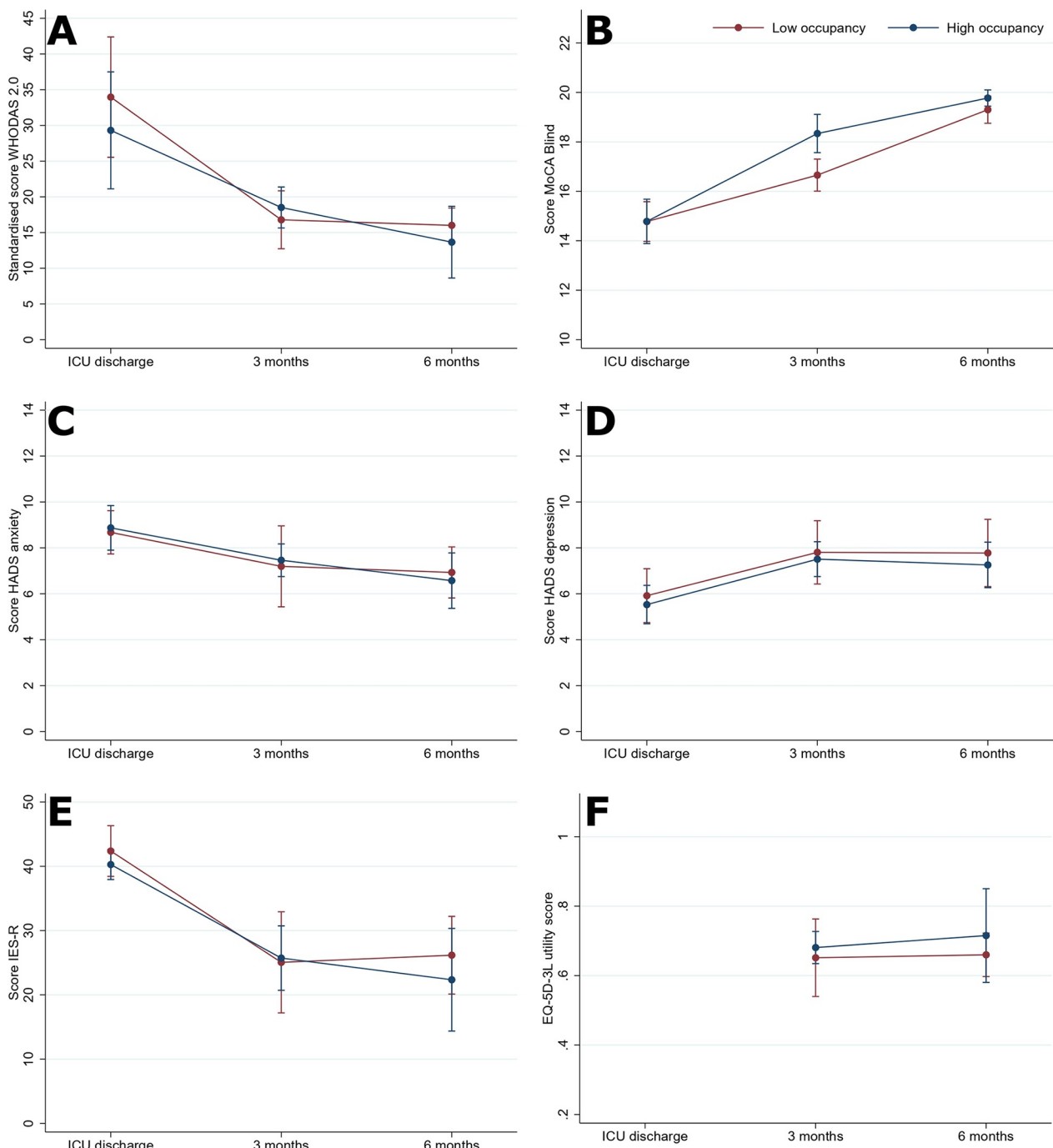

**Fig 2. Mental, physical, and cognitive impairments of ICU survivors by bed occupancy were estimated with a multilevel linear regression model adjusted for age, sex, educational level, and COVID-19 infection status.** A: Standardized World Health Organization Disability Assessment Schedule (WHODAS 2.0) score–disability level; B: Montreal Cognitive Assessment (MoCA)-Blind score–cognitive function; C: Hospital Anxiety and Depression Scale (HADS) score- anxiety subscale; D: HADS score- depression subscale; E: Impact of Events Scale-Revised (IES-R)-post-traumatic stress symptoms; F: EQ-5D-3L utility score- health-related quality of life.

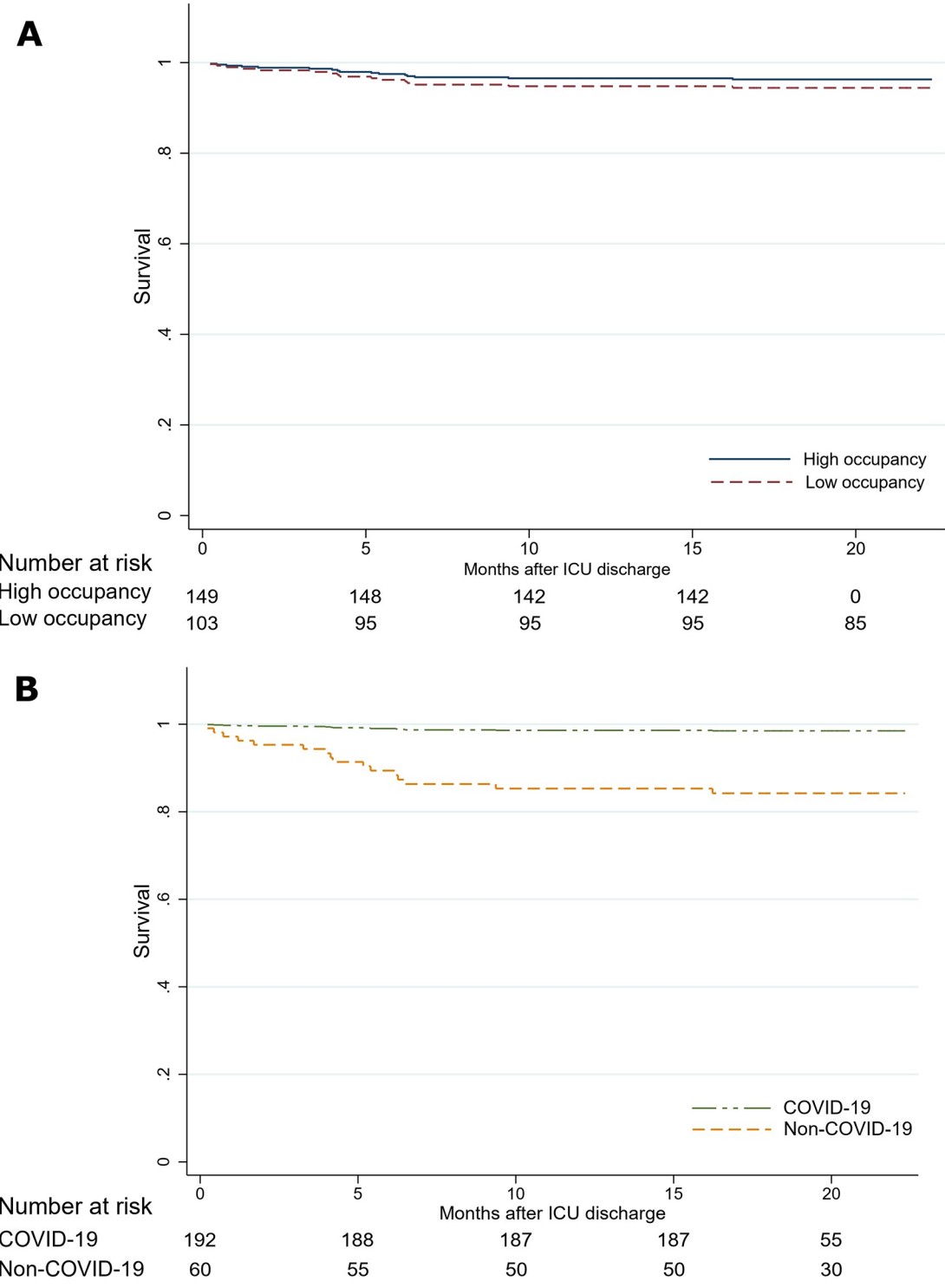

**Fig 3. Cox regression model adjusted for age, sex, and educational level.** A: by occupancy level; B: by COVID-19 infection status.

## Discussion

This is the first multicenter longitudinal study of post-ICU outcomes in Chile, assessing patients at ICU discharge and comparing mental, physical, and cognitive impairments during high and low bed occupancy periods. We found that bed occupancy and infection with COVID-19 are not associated with the mental, physical, and cognitive impairments ICU survivors present at ICU discharge, 3- and 6-month follow-up. Most patients had at least one impairment at ICU discharge (92%), which decreased to 68% at the 3-month and 57% at the 6-month follow-up.

Outcomes of patients treated in periods of low bed occupancy were similar to those treated during high occupancy. In principle, high bed occupancy could lead to poorer quality of care because staff-to-patient ratios are lower [29], staff work under greater pressure, increasing the chance of making mistakes [30], dismissing early warning signs, and paying less attention to care that is considered lower priority [31]. The pandemic could be seen as a natural experiment allowing the impact of high bed occupancy on mortality and other patient outcomes to be measured, particularly if interventions that reduce or prevent PICS are perceived as lower priority. The fact that we did not find differences in the frequency of PICS or mortality might indicate that bed occupancy was not associated with quality of care, or, at least, it might not have modified quality of care to the extent of affecting patients' functional outcomes. However, bed occupancy was measured at a national level; therefore, it might not reflect bed occupancy at each study site. Additionally, this study started recruiting patients six months after the COVID-19 pandemic was declared, which might have given clinical teams time to adapt to a higher workload.

The demographic profile of patients admitted was different between occupancy periods. We attribute this to how COVID-19 spread, infecting first the most vulnerable population (i.e., older, less educated with more long-term chronic conditions) and then infecting healthier patients. However, despite these differences, outcomes were similar. It is noteworthy that patients admitted with other non-COVID-19 diagnoses were frailer, had more long-term chronic conditions, and were older, indicating that ICU admission thresholds might have been more stringent for non-COVID-19 patients during the pandemic. It is also plausible that these patients might have been admitted late, which combined with a higher clinical severity, could partially explain the lower survival for patients admitted due to other diagnoses during the pandemic.

The frequency of at least one impairment was 92% at ICU discharge. There are no comparable studies with an assessment at ICU discharge during the pandemic; therefore, it is not possible to make a judgement whether this rate is high or not.

Nevertheless, we can compare some of our findings to studies conducted before the pandemic. For example, regarding physical impairments at ICU discharge, 34% of patients had ICU-AW, which is similar to levels reported in the pre-pandemic literature [32]. Similarly, our population median for the Functional Status Scale for the ICU was similar to pre-pandemic studies conducted in Chile [33] and elsewhere [34].

Regarding mental and cognitive impairments, a prior study [35] found lower rates of cognitive impairments (58% vs 72%), anxiety (47% vs 58%), and symptoms of post-traumatic stress (37% vs 65%) at ICU discharge, with a similar pattern at the 3-month follow-up. These differences could be explained by the clinical characteristics of the population assessed, where all our patients were mechanically ventilated with a median duration of seven days and a median ICU stay of 15 days, while this other prospective study [35] reported 46% of patients receiving mechanical ventilation, and a median ICU stay of 4.2 days.

In comparison with the most recent systematic reviews [36–38], we found a similar rate of anxiety (37% [28 to 48] vs 30%[19 to 41]) and depression (42% [33 to 52] vs 24%[17 to 32]) but a higher rate of post-traumatic stress symptoms (35% [26 to 45] vs 13%[2 to 23]) at the 3-month follow-up. Early post-ICU memories of frightening experiences and post-ICU psychopathology are reported as significant risk factors for post-traumatic stress symptoms [39], which might explain our high rates.

In the case of cognitive impairments, our rate of 36% [27 to 46] at 3-month follow-up lies between the two more recent systematic reviews [38, 40], and it is very similar to an ambispective cohort study of 186 mechanically ventilated COVID-19 patients[4]. We found similar rates of cognitive impairments at the 6-month follow-up to those already reported, although these studies have recruited patients with varying levels of severity and have used different instruments for measuring cognitive impairments. Future research should measure post-ICU impairments consistently to improve understanding of which factors affect the frequency and severity of PICS.

At 6-month post-discharge, similar studies assessing mental, physical, and cognitive impairments during the pandemic [41] report higher rates of PICS (80% vs 56%) but a similar frequency of at least one mental impairment (50% vs 54%). However, pre-pandemic studies [42, 43] report much lower mental impairment rates (14.6% and 22%, respectively). The frequency of anxiety in our study is similar to previous ones [36], while depression is similar but on the higher end; however, cases of post-traumatic stress are much higher, even compared with Hodgson et al. [44], which included a cohort of COVID-19 patients. Future research should focus on identifying patients at greater risk of presenting mental impairments and the extent to which the environment in the ICU during the pandemic might have caused more mental impairments in those who survived, which is already being reported [45].

Finally, we found similar rates of disability, return to work, and quality of life levels to those reported before the pandemic [5, 42, 43], and those reported for COVID-19 patients [44], which shows that patients admitted to ICU during the pandemic have similar impairments to ICU survivors treated before 2020. Therefore, healthcare systems should provide appropriate rehabilitation to patients at higher risk of developing impairments, such as patients with known mental, physical, and cognitive impairments and those with long-term chronic conditions [46]. The volume of patients seen during the pandemic has helped to shine a light on what it means to live after surviving intensive care and should be seen as an opportunity to improve post-ICU services.

This study has some limitations. The attrition rate was higher than comparable follow-up studies measuring disability; however, our respondents' clinical and demographic characteristics were not different from the sample we initially recruited; therefore, we have confidence that our estimates are robust. Due to the restrictions on social interactions in the country, we had limited options to assess physical functioning; therefore, we decided to conduct a comprehensive assessment at ICU discharge and an extensive telephone follow-up. Lastly, we excluded patients who were uncooperative or delirious at ICU discharge because they could not follow complex instructions, or their answers might be unreliable; however, these patients might have had worse mental, physical, and cognitive outcomes, and therefore, if we had included them, our estimates might be higher.

In conclusion, 92% of patients had at least one mental, physical or cognitive impairment at ICU discharge, which decreased to 57% at 6-month follow-up. Bed occupancy and COVID-19 infection were not associated with post-ICU patient's functional outcomes. The frequency of mental, physical, and cognitive impairments of patients treated during the pandemic was similar to other studies, except for the frequency of post-traumatic stress symptoms, which was much higher. Future studies should focus on how to increase ICU capacity in emergencies without affecting the quality of care.

## Supporting information

**S1 Fig. National ICU beds occupancy levels between May 1st 2020 and April 30th 2021.**
Data retrieved from www.minciencia.gob.cl/COVID-19 and plotted by the authors.
(TIFF)

**S1 Table. Description of the measurement instruments used in the IMPACCT COVID-19 Study.**
(DOCX)

**S2 Table. Functional outcomes and employment status.**
(DOCX)

**S3 Table. Baseline characteristics and intensive care unit outcomes at ICU discharge of patients assessed at 3 and 6 months.**
(DOCX)

**S4 Table. Functional outcomes and employment status at intensive care unit discharge of patients assessed at 3 and 6 months and those lost-to-follow-up.**
(DOCX)

**S5 Table. Health related quality of life and employment status at intensive care unit discharge, 3 months and 6 months follow-up according to bed occupancy.**
(DOCX)

**S6 Table. Functional outcomes and employment status at intensive care unit discharge, 3 months and 6 months follow-up according to COVID-19 infection.**
(DOCX)

**S7 Table. Functional outcomes at intensive care unit discharge according to bed occupancy and COVID-19 infection.**
(DOCX)

**S8 Table. Functional outcomes and employment status at 3 months and 6 months follow-up according to COVID-19 infection and bed occupancy.**
(DOCX)

**S9 Table. IMPACCT COVID-19 study group sites and evaluators.**
(DOCX)

**S1 Checklist. STROBE statement—Checklist of items that should be included in reports of *cohort studies.***
(DOCX)

## Acknowledgments

We want to acknowledge each participating site for allowing us to use part of their infrastructure (organized alphabetically): *Clínica Alemana de Santiago*, *Clínica BUPA*, *Clínica INDISA*, *Hospital del Salvador*, *Hospital Metropolitano*, *Hospital Regional Dr Leonardo Guzmán de Antofagasta*, and *Hospital San Pablo de Coquimbo*. We also acknowledge Prof Karen Bloor (University of York), Prof Tim Doran (University of York), and Dr Kirby P. Mayer (University of Kentucky) for their comments on earlier versions of this manuscript.

Collaborators:

The IMPACCT COVID-19 study group includes (organized alphabetically): Ana Castro-Avila, Agustín Camus-Molina, Catalina Merino-Osorio, Felipe González-Seguel, and Jaime

Leppe; and the following additional collaborators: Camilo Cáceres-Parra, Eduardo González Tapia, Felipe Muñoz-Muñoz, Fernanda Baus Auil, Javiera Aguilera Scarpati, Joaquín Olave, Macarena Leiva-Corvalán, Pilar Castro, and Yerko Villagra Jofré. See the full list of all assessors and participating sites in the Supporting Information (S9 Table).

## Author Contributions

**Conceptualization:** Ana Castro-Avila, Catalina Merino-Osorio, Felipe González-Seguel, Agustín Camus-Molina, Jaime Leppe.

**Data curation:** Ana Castro-Avila, Catalina Merino-Osorio, Felipe González-Seguel, Agustín Camus-Molina.

**Formal analysis:** Ana Castro-Avila.

**Funding acquisition:** Ana Castro-Avila, Catalina Merino-Osorio.

**Investigation:** Ana Castro-Avila, Catalina Merino-Osorio, Felipe González-Seguel.

**Methodology:** Ana Castro-Avila, Catalina Merino-Osorio, Felipe González-Seguel, Agustín Camus-Molina, Jaime Leppe.

**Project administration:** Ana Castro-Avila, Catalina Merino-Osorio, Felipe González-Seguel, Felipe Muñoz-Muñoz.

**Resources:** Ana Castro-Avila, Jaime Leppe.

**Supervision:** Ana Castro-Avila, Catalina Merino-Osorio, Felipe González-Seguel, Agustín Camus-Molina, Felipe Muñoz-Muñoz.

**Validation:** Ana Castro-Avila.

**Writing – original draft:** Ana Castro-Avila, Catalina Merino-Osorio, Felipe González-Seguel, Agustín Camus-Molina, Felipe Muñoz-Muñoz, Jaime Leppe.

**Writing – review & editing:** Ana Castro-Avila, Felipe González-Seguel, Felipe Muñoz-Muñoz.

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
