## [Decision Letter · Decision Letter 0]

17 Oct 2023

PONE-D-23-23670Six-Month Post-Intensive Care Outcomes During High and Low Bed Occupancy due to the COVID-19 Pandemic: A Multicenter Prospective Cohort StudyPLOS ONE

Dear Dr. González-Seguel,

Thank you for submitting your manuscript to PLOS ONE. After careful consideration, we feel that it has merit but does not fully meet PLOS ONE’s publication criteria as it currently stands. Therefore, we invite you to submit a revised version of the manuscript that addresses the points raised during the review process.

ACADEMIC EDITOR: Thank you for submitting your manuscript to PLOS ONE. We reviewed the manuscript carefully and decided to minor revision. Therefore, we invite you to submit a revised version of the manuscript that addresses the points raised during the review process.==============================

We look forward to receiving your revised manuscript.

Kind regards,

Ahmet Çağlar, M.D.

Academic Editor

PLOS ONE

“We want to acknowledge the Universidad del Desarrollo for funding this research and each participating site for allowing us to use part of their infrastructure (organized alphabetically): Clínica Alemana de Santiago, Clínica BUPA, Clínica INDISA, Hospital del Salvador, Hospital Metropolitano, Hospital Regional Dr. Leonardo Guzmán de Antofagasta, and Hospital San Pablo de Coquimbo.”

“IMPACCT COVID-19 study was funded by Universidad del Desarrollo (Grant number

2020-78) and sponsored by the Chilean National Agency for Research and

Development (ANID-0772). The funders had no role in the design, collection, analysis

of the study, and writing of this manuscript. Received by CM-O.”

Reviewers' comments:

Reviewer's Responses to Questions

**Comments to the Author**

1. Is the manuscript technically sound, and do the data support the conclusions?

Reviewer #1: Yes

2. Has the statistical analysis been performed appropriately and rigorously? 

Reviewer #1: Yes

3. Have the authors made all data underlying the findings in their manuscript fully available?

Reviewer #1: Yes

4. Is the manuscript presented in an intelligible fashion and written in standard English?

Reviewer #1: Yes

5. Review Comments to the Author

Reviewer #1: Congratulations on your hard work.

The word ‘per cent’ in methods and findings section of abstract must be corrected.

I found a few writing errors in all text. Other than that, I found the article carefully prepared and well-done.

6. PLOS authors have the option to publish the peer review history of their article (what does this mean?). If published, this will include your full peer review and any attached files.

Reviewer #1: No

---

## [Author Response · Author response to Decision Letter 0]

23 Oct 2023

Please, see the attached letter with all responses together.

REVIEWER #1: 

Congratulations on your hard work. The word ‘per cent’ in methods and findings section of abstract must be corrected. I found a few writing errors in all text. Other than that, I found the article carefully prepared and well-done.

ANSWER: We appreciate your review. We have corrected that writing error in the abstract and we checked for typos in the rest of the text.

---

## [Editor Report · Decision Letter 1]

5 Nov 2023

Six-Month Post-Intensive Care Outcomes During High and Low Bed Occupancy due to the COVID-19 Pandemic: A Multicenter Prospective Cohort Study

PONE-D-23-23670R1

Dear Dr. González-Seguel,

We’re pleased to inform you that your manuscript has been judged scientifically suitable for publication and will be formally accepted for publication once it meets all outstanding technical requirements.

Kind regards,

Ahmet Çağlar, M.D.

Academic Editor

PLOS ONE

Additional Editor Comments (optional):

Dear Felipe González-Seguel

I have reviewed your revised manuscript, and I am pleased to report that the manuscript is now acceptable for publication.
---

## [Editor Report · Acceptance letter]

9 Nov 2023

PONE-D-23-23670R1 

Six-month post-intensive care outcomes during high and low bed occupancy due to the COVID-19 pandemic: a multicenter prospective cohort study 

Dear Dr. Castro-Avila:

I'm pleased to inform you that your manuscript has been deemed suitable for publication in PLOS ONE. Congratulations! Your manuscript is now with our production department. 

Kind regards, 

on behalf of

Dr. Ahmet Çağlar 

Academic Editor

PLOS ONE